# Effects of Forest Trail and Ground Walking on Mental and Physical Health Promotion in Middle-Aged Women Living in Urban Areas

**DOI:** 10.3390/healthcare13222876

**Published:** 2025-11-12

**Authors:** Eunheui Nam, Seongwoo Jeon

**Affiliations:** 1Department of Environmental Science and Ecological Engineering, Graduate School, Korea University, Seoul 02841, Republic of Korea; nam7195@naver.com; 2Department of Environmental Science and Ecological Engineering, College of Life Sciences and Biotechnology, Korea University, Seoul 02841, Republic of Korea

**Keywords:** electroencephalography (EEG), heart rate (HR), urban planning policy, public health policy, wearable device

## Abstract

**Background/Objectives:** Recently, the importance of physical activity for health promotion has increased the demand for physical activities performed in natural environments. However, environmental characteristics that enhance the efficiency of physical activities and contribute to health promotion have not yet been established. This study aimed to verify the mental and physical health of walking in different environments by measuring EEG and HR responses among middle-aged women living in urban areas during forest trail (GU) and school ground (NF) walking. **Methods:** In total, 30 middle-aged women participated in a 1.5 km walking, with HR measured during normal, NF, and GU walking. EEGs were recorded before and after walking 5 waves (Delta, Theta, Alpha, Beta, and Gamma). All data were collected under standardized conditions and analyzed using paired t-tests. **Results:** Alpha, beta, and gamma waves increased after GU walking (*p* < 0.001) but decreased after NF walking, suggesting that walking in natural environments promotes emotional stability, attentional recovery, and cognitive activation. Mean HR during GU was higher than during NF (*p* < 0.001), and NF walking corresponded to moderate-intensity exercise, whereas GU walking represented vigorous-intensity activity, likely influenced by its 5% slope and multi-sensory natural stimuli such as forest, sounds, and air quality. **Conclusions:** This study is not a clinical trial but a health experiment of physical activity, highlighting how walking in natural environments can contribute to improved health. The walking environment elicits distinct mental and physical responses, and forest walking has proven to be more effective in improving health. This result highlights the value of nature-based exercise spaces accessible in urban environments and can help with design and health policies.

## 1. Introduction

Recently, the importance of physical activity for promoting mental and physical health has been emphasized worldwide [1], and the significance of natural environments in enhancing overall health has been recognized [2]. The positive effect of being in contact with nature on health has been discussed across various domains, and its effectiveness has been demonstrated [3,4]. A brief view of forest landscapes induces mental relaxation [5], and physical activity in natural spaces has been reported to be more effective in improving cardiovascular and mental health than that in urban environments without nature [6]. Recently, meta-analyses have indicated that walking in natural environments is more effective than walking in urban environments for reducing the risks of depression and anxiety, facilitating heart rate recovery, alleviating fatigue, and enhancing vitality [7].

Light exercise in natural environments has shown better results in regulating blood pressure, increasing self-esteem, and enhancing positive mood compared to indoor settings [8,9]. Forest bathing has been found to offer various physiological and psychological benefits, including stress reduction, decreased heart rate, lower cortisol levels, and enhanced immune function [10,11]. Activities in forests have significantly improved the physical and mental health of individuals with chronic diseases, such as breast cancer, chronic obstructive pulmonary disease (COPD), and cardiovascular conditions.

Furthermore, walking in natural environments induces prefrontal activation and increases alpha wave activity, thereby promoting mental relaxation and enhancing cognitive function [12,13].

According to the WHO, health is defined as a state of complete physical, mental, and social well-being [1]. The environment can be understood as a relational concept between the subject and the object, in which humans act as primary agents interacting with environmental elements such as air, water, and sound. Although environmental values have traditionally been assessed using measurable indicators such as air quality, water pollution, and noise levels, such approaches are limited in capturing human-centered aspects of health [14].

Based on the WHO’s comprehensive concept of health, this study focused on two key components—mental and physical health—within different environmental contexts. By positioning humans as central subjects within their environments, this study aimed to analyze how varying environmental conditions influence these dimensions of health and to identify environmental characteristics that contribute to health promotion.

However, although previous studies have primarily focused on well-being and healing, there remains a lack of research that quantitatively verifies the effects of nature-based physical activity in daily life on mental and physical health indicators [15,16].

To address this knowledge gap, this study aimed to quantitatively verify the effects of physical activities, such as walking in natural environments, on mental and physical health. For this purpose, middle-aged women who are sensitive to environmental changes and frequently engage in walking activities [17] were selected as participants.

The EEG was selected, as it provides objective data that quantifies emotional states conveniently and reliably [18], while heart rate (HR) is widely used as an indicator of exercise intensity and is effective in indirectly analyzing the effects of exercise [19]. Changes in mental state were quantitatively measured using EEG [20,21,22,23], and HR was recorded to assess differences in physical exercise intensity [19,24,25].

We then analyzed the effects of physical activity based on these indicators. Furthermore, two different walking environments—forest trails and school playgrounds—were compared to quantitatively determine how environmental differences affect emotional stability, attention recovery, and exercise intensity in middle-aged women.

Ultimately, this study aims to identify optimal physical activity spaces and provide empirical evidence for developing effective health promotion strategies to enhance the well-being of middle-aged women living in urban areas.

## 2. Materials and Methods

### 2.1. Study Area

To verify the health-promoting effects of physical activity environments, various physical activity settings could be considered, including indoor environments such as gymnasiums or fitness studios. However, this study focused solely on outdoor environments to enable a direct comparison with natural environments. Therefore, this study focused on urban outdoor environments, which middle-aged women find easily accessible in their daily lives, prioritizing the examination of differences in the effects of exercise based on the presence of forests and slopes (Table 1).

First of all, a forested area with a gentle slope (GU) and a non-forested flat area (NF) were selected for comparative analysis (Table 1). Therefore, in this study, NF can be regarded as a comparative condition, that is, a functional control for GU.

The GU site is a barrier-free trail located at the entrance of Umyeonsan Mountain (maximum elevation 293 m) in the central area of Seocho-gu, Seoul. Surrounded by residential and commercial districts, it is an urban mountain that local residents can easily access on foot or by public transportation and is widely used as a space for daily walking and health activities. It offers excellent accessibility, allowing local residents of all ages to reach and enjoy the site easily. The trail has an average gentle slope of around 5%, providing a safe walking path that anyone, including the elderly and other mobility-impaired, vulnerable groups, can use comfortably, making it a representative urban natural area.

The NF site is a school playground at Seoul High School, characterized by its few natural elements and flat terrain. Both sites, located within 1 km of each other, are accessible, frequently used public spaces with distinctly contrasting features, making them suitable for analyzing the impact of environmental characteristics on changes in mental and physical health among participants undertaking walking activities (Figure 1).

### 2.2. Participants

Middle-aged women, who, according to the WHO’s definition, were aged between 40 and 59 years, were highly responsive to environmental stimuli during walking activities on trails, and emotionally sensitive around the menopausal period [17], were selected. Compared with men, middle-aged women tend to rely more on walking for physical activity [26], making them appropriate participants for this study.

Based on prior research and to ensure statistical reliability, a sample size of 30 women was established as the optimal experimental group size. Participants were recruited through an open call using public flyers, and a non-probability purposive sampling method was applied. This study was not a medical clinical trial or an intervention targeting specific treatments or diseases, but rather an exploration of how mental and physical responses vary according to walking activities in daily life. Therefore, the inclusion criteria specified women aged 40–59 years without any special restrictions related to medical conditions. However, individuals who ① had physical disabilities or ② had difficulty walking more than 2 km were excluded.

Participants filled out an online questionnaire using the Google form before walking. The pre-walk survey collected data on demographic characteristics (e.g., age and area of residence), preexisting medical conditions, habitual physical activity levels (frequency per week and average hours), and perceived stress (frequency in the last two weeks).

The final sample consisted of 30 healthy women with a mean age of 51.2 years. None of the participants had physical or mental disorders, and none were taking medications. On average, they reported walking for more than one hour per day on approximately 5.2 days per week, whereas moderate or vigorous exercise was performed only about twice per week. All participants were middle-aged women living in healthy metropolitan areas who performed walking activities regularly and had relatively low mental stress (Table 2). All participants successfully completed walking experiments in both environments (school ground and forest trail). Regarding the post-walking survey, it was excluded from the main analysis because the primary aim of this study was quantitative measurement rather than subjective perception.

This study was approved by the Institutional Review Board of Korea University (KUIRB-2025-000501). All participants were fully informed of the study’s objectives and procedures and provided written informed consent prior to participation.

### 2.3. Methods

#### 2.3.1. Methods of Measurement

The WHO describes walking as the most accessible and universal form of physical activity [1], which acts as an effective means for promoting physical and mental health among middle-aged women [27]. To quantitatively assess the health-promoting effects of walking, this study measured changes in mental health using an EEG and physical health using a heart rate monitor.

The measurement procedures were conducted in two sessions—first on the school ground and then on the forest trail (Figure 2)—and all procedures were carried out by a research team consisting of three trained research assistants and the principal investigator. The research assistants, who had received prior training in the use of physiological monitoring instruments for physical activity assessment, were responsible for EEG device setup and for measuring heart rate. All devices underwent calibration procedures prior to the experiment to ensure signal accuracy. For the EEG device (Muse 2), a trained research assistant performed calibration and attachment after Bluetooth synchronization, checking electrode contact status, reference electrode stability, and external noise to maintain a stable signal. Before heart rate measurement, the assistant verified the wearing position and Bluetooth synchronization of the HR device (Polar) in advance, and each participant monitored signal stability through their individual device while walking. The principal investigator supervised the entire process to ensure accuracy and consistency in data collection.

The experiment was conducted using random assignments generated by a computer-based random number generator. Each participant performed the walking sessions individually to prevent group influence and ensure consistent individual measurements. After completing the walking session on the school ground track, participants were transported by vehicle to the forest trail site and performed the second walking session after a rest period (Figure 2).

All experiments were conducted under comparable weather conditions (clear sky, wind speed below 3 m/s) and within a consistent temperature range (18–20 °C, 10:00 a.m.–5:00 p.m.), and all sessions were performed within the same week to ensure comparable environmental conditions and to minimize environmental influences. The researcher accompanied all sessions to ensure participant safety and to monitor the measurement procedures.

#### 2.3.2. Mental Health Measurement

In this study, EEG measurements were conducted to collect objective and quantitative data on the psychological effects of outdoor activity.

Because EEG signals measured outdoors can be influenced by various factors, all participants were allowed to rest and stabilize before the experiment.

As this study aimed to explore public health and well-being in daily life rather than to conduct a clinical or medical experiment, a Muse 2 mobile EEG device (InteraXon, Canada), a headband-type EEG device, was used. This device is easy to wear in outdoor environments, allows real-time monitoring of brainwave changes via a smartphone application, and enables convenient data collection. Therefore, it is convenient and portable for outdoor measurements. EEG data were collected for 5 min before and after walking on both the school ground and forest trail, resulting in 4 measurements per participant. The frequency bands analyzed included Delta, Theta, Alpha, Beta, and Gamma waves, and electrodes were placed at TP9, AF7, AF8, and TP10 (Figure 3). 

EEG electrode attachment was performed uniformly for all participants by trained research assistants. During the measurements, participants sat comfortably in a chair with a backrest, closed their eyes, and remained still to minimize motion artifacts. Although complete isolation from external factors was not possible in the outdoor environment, EEG recordings were conducted in quiet and controlled settings to ensure data reliability.

#### 2.3.3. Physical Health Measurement

In this study, heart rate was measured as a simple and stable indicator that allows for the quantitative evaluation of differences in exercise intensity between environments and the assessment of physical health changes.

Heart rate was measured using a Polar H10 chest-strap heart rate monitor (Polar Electro, Finland). Prior to attachment, the skin contact area on the chest was disinfected with an alcohol swab to ensure accurate signal detection. All participants, under the guidance of trained research assistants, securely attached the sensor at the center of the chest. All procedures were conducted under the supervision of the principal investigator.

To assess the normal heart rate, participants rested upon arrival at the experimental site and then sat in a chair with a backrest, maintaining a stable posture for 5 min while their heart rate was recorded. Subsequently, participants wore the heart rate monitor and performed 1.5 km walking sessions on the school playground and forest trail, in that order. Participants equipped with wearable HR monitors were instructed to walk on the flat school ground (0–2% slope) at a pace slightly faster than their normal speed, inducing mild breathlessness. On the forest trail, which had an approximate 5% slope, participants were asked to walk at a pace that felt similar to the speed maintained on the school ground. Heart rate data were continuously recorded in real time via a mobile application from the start to the end of each walking session, resulting in 3 sets of heart rate data per participant (normal, during track walking, during forest walking).

### 2.4. Analysis

All statistical analyses were performed using the Data Analysis Tool Pak in Microsoft Excel. First, an independent sample t-test was conducted to compare the pre- and post-walking differences within each environment (ground and forest trail). Subsequently, a one-way analysis of variance (ANOVA) was performed to examine the overall differences among the four episodes (before and after walking on the playground and before and after walking on the forest trail). For all variables, the mean (M) and standard deviation (SD), Variance (Var) were calculated, and the level of significance was set at *p* < 0.05.

## 3. Results and Discussion

### 3.1. EEG Results

EEG data were collected from 30 middle-aged women across four episodes—before and after walking the school grounds (episodes 1 and 2) and before and after walking the forest trail (episodes 3 and 4). This study analyzed changes in brainwave activity across these episodes in relation to the functional brain areas associated with the electrode sites.

#### 3.1.1. Pre- and Post-Walking EEG Changes

To examine changes in brainwave activity across the four measurement sessions—before and after walking on the school ground and forest trail—data for five brainwave types (Delta, Theta, Alpha, Beta, and Gamma) were analyzed (Figure 3). Each box plot displays the median, maximum, minimum, and variance values to visualize the distribution patterns, and subsequent analyses were performed to compare changes between the two environments. Figure 4 visually compares the distributions of five types of brain waves across the four measurement episodes. The box plots represent the median values and the distribution of normalized EEG data. Overall, after ground walking (Ep. 2), all brainwave values appeared more dispersed, with Delta and Theta waves showing little change but a slight decrease.

The median value of Alpha waves slightly decreased, and both Beta and Gamma waves showed a downward trend in their median and mean values. This suggests that walking in an artificial playground environment had limited effects on emotional stability and cognitive activation. In contrast, a clear difference was observed between forest walking before and after (Ep. 3 and 4). Alpha, Beta, and Gamma waves all significantly increased after forest walking, with a particularly notable rise in the median value of Alpha waves. Theta waves also showed a slight increase, suggesting that the forest environment may have promoted emotional stability and a relaxed yet alert state. When comparing Episode 2 and Episode 4, the median values of all EEGs were higher after forest walking, with Alpha, Beta, and Gamma waves showing the most prominent differences. Additionally, the upper range of the box plots expanded overall, indicating enhanced psychological relaxation, attentional recovery, and improved cognitive activity levels following forest walking.

Delta waves showed no significant differences between before and after ground walking or before and after forest trail walking (*p* > 0.05), and Theta waves exhibited no notable changes across sessions (Figure 5). This suggests that walking did not alter brainwave levels typically associated with deep relaxation, sleep, or meditative states. In contrast, the average values of Alpha, Beta, and Gamma waves significantly decreased after ground walking compared with before ground walking. In contrast, they significantly increased after forest trail walking compared with before forest trail walking, with strong statistical significance (*p* < 0.001). These findings suggest that forest walking positively influences various aspects of mental health, including emotional stability, attention, and cognitive activation, whereas walking in a school playground environment does not yield such mental health benefits.

The increase in Alpha waves, associated with psychological relaxation and stress recovery [28], after forest walking indicates the effectiveness of natural stimuli in promoting emotional stability. The increase in Beta waves, which are related to attention and arousal [29], suggests that sensory stimuli from the forest walking may enhance alertness and focus. The increase in Gamma waves, which are associated with higher-order cognitive functions, memory, learning, and creative thinking [30], indicates that forest walking positively influences cognitive activation. In contrast, decreased EEG activity after ground walking aligns with previous findings that artificial urban environments may impose a cognitive burden, rather than facilitating cognitive recovery [31].

These findings suggest that simple physical activity alone may be insufficient for achieving emotional and cognitive recovery, and that urban or artificial environments, such as school playgrounds, may even place additional burdens on mental health. Furthermore, the results indicate that the effects of walking are more sensitive to the qualitative characteristics of the environment than to the amount or duration of physical activity. In particular, walking in forest environments appears to promote mental health among middle-aged women living in urban areas by providing emotional stability and positive cognitive stimulation.

#### 3.1.2. EEG Changes by Electrode Site

EEG was performed at TP9, AF7, AF8, and TP10 according to the International 10–20 System [32], with each site corresponding to a specific brain function (Figure 6). TP9 and TP10, which are located in the temporal-occipital regions, are associated with auditory processing, emotional recall, and spatial cognition [33,34], whereas AF7 and AF8, which are located in the anterior frontal regions, are associated with attention, cognitive activities, emotional responses, and stress regulation [35,36].

The radar chart showed consistently higher brainwave activity in areas related to emotion, attention, and cognition, particularly at Delta_TP9, Alpha_AF7, and TP10, reflecting sensitivity to environmental changes. After walking on the ground, slight decreases or no changes in the Alpha and Theta waves were observed, indicating limited effects on emotional, attentional, and relaxation functions (Table 3). 

In contrast, the Alpha, Theta, and Gamma waves significantly increased across all sites after walking in the forest, demonstrating that the natural environment simultaneously facilitated emotional stability, attention recovery, and cognitive activation [13,31]. These findings indicate that forest walking environments can have a positive physiological impact on brain function in middle-aged women, who are particularly responsive to natural environments and have strong emotional associations with physical activity [37].

### 3.2. Heart Rate Results

The heart rates of 30 middle-aged women participants were measured before walking (in a normal state), during ground walking, and during forest trail walking to examine the benefits of walking on physical health in accessible urban spaces.

#### 3.2.1. Heart Rate Statistical Analysis

An ANOVA test was conducted to examine changes in heart rate among middle-aged women participants under three conditions: before walking (B), ground walking (G), and forest trail walking (F). As shown in Table 4, the mean heart rates were 73.78 bpm at the normal state (before walking [B]), 120.69 bpm during F, and 114.78 bpm during G. The *p*-value was <0.001, indicating statistically significant differences between conditions (Table 4).

Comparisons with B showed that G increased to 41.00 bpm, whereas F led to an increase in 46.91 bpm, demonstrating that both environments induced significant increases in heart rate (*p* < 0.001). Additionally, when comparing the two environments, F showed a higher heart rate of 5.91 bpm than that of G (*p* < 0.001). These findings confirm that physical activity in a forest environment consistently induces higher heart rate levels.

#### 3.2.2. Exercise Intensity Analysis

Using the Fox & Haskell Formula (HRmax = 220 − age), the estimated maximum heart rate in participants with an average age of 51.2 years was 170 bpm. The exercise intensity analysis showed that the normal heart rate was equivalent to the resting level, G corresponded to moderate-intensity aerobic exercise, and F corresponded to high-intensity aerobic exercise (Table 5).

Walking on sloped forest trails increased the heart rate and required higher exercise intensity than walking on flat tracks; the average HR recorded during F was 5.91 bpm higher than G, indicating that environmental factors substantially influenced exercise intensity [38].

#### 3.2.3. Time-Series Analysis of Heart Rate Changes

To examine the changes in HR during F and G walking, a time-series analysis was conducted using the minute-by-minute average HR data. This analysis allowed for a visual comparison of HR variability and differences in exercise intensity between the two conditions. The average speed recorded for G walking was 10.01 min/km, whereas that for F walking was 12.59 min/km. The difference in walking speed was due to the approximately 5% slope of the forest trail. The higher heart rate observed during forest walking may have resulted from the increased physical demand required to walk uphill on the sloped terrain (approximately 5%), which in turn could have influenced participants’ physiological responses. These findings indicate that, for middle-aged women, F walking with a gentle slope of approximately 5% can contribute to increased exercise intensity and total workload (Figure 7).

Therefore, F walking can be considered more effective in improving cardiorespiratory endurance and providing greater physical stimulation than G walking. Furthermore, the error bar graph indicates that HR variability tends to remain relatively stable during F walking. This difference can be interpreted not merely because of differences in distance or time but also as a reflection of the psychological and physiological stimuli induced by terrain inclines and natural elements. The gentle, continuous slope of F may have provided a balanced exercise stimulus. In contrast, the natural environment of the forest likely offered cognitive immersion and emotional stability, allowing participants to focus more consciously on the activity. This, in turn, may have naturally maintained or increased the exercise intensity, which aligns with previous findings that natural environments can enhance exercise adherence and intensity by providing psychological benefits [39].

### 3.3. Discussion

This study analyzed the effects of walking environments on EEG activity and heart rate among middle-aged women living in urban areas.

However, it may not be scientifically sufficient to conclude that psychological well-being improved solely based on EEG changes. Although the observed EEG variations may have resulted from exposure to the natural environment, it cannot be entirely ruled out that situational factors during the experiment also influenced the results. Moreover, while the portable EEG device used in this study offers the advantage of applicability in outdoor environments, unlike laboratory settings, external factors such as weather conditions, illumination, and ambient noise may have subtly affected the measurements. These characteristics represent an inherent limitation of field studies conducted in natural environments; therefore, future research should include supplementary measurements under controlled laboratory conditions.

Nevertheless, the findings of this study were consistent with previous research in two main respects. First, increases in alpha, beta, and gamma waves have been repeatedly reported in neurophysiological studies to be associated with emotional stability, psychological relaxation, and cognitive activation [40]. Second, the observed increase in alpha wave activity in the temporo-parietal (TP) region aligns with previous research [41], which found a close relationship between enhanced TP alpha activity, psychological well-being, and emotional regulation functions.

The increase in alpha waves observed after forest walking indicates relaxation and attentional restoration, while the increases in beta and gamma waves reflect enhanced cognitive processing and integrative thinking. These findings suggest that multisensory stimuli such as the forest’s visual greenery, natural sounds, and fresh air may promote both neural stability and cognitive activation.

However, as this study has the characteristics of a pilot study, it has limitations in drawing definitive causal inferences. Therefore, future research should incorporate psychological scales (e.g., POMS, PANAS) and cognitive tasks (e.g., working memory tests) to comprehensively analyze both subjective and objective indicators, and conduct longitudinal monitoring to verify the persistence of these effects.

In terms of physiological responses, the mean heart rate during forest walking was slightly higher than that during track walking. This difference may be attributed to the approximately 5% slope and uneven terrain of the forest trail. Environmental factors such as phytoncides, negative ions, and natural sounds may also have contributed to maintaining a stable increase in heart rate and improving energy efficiency.

Regarding exercise intensity, when applying the Fox and Haskell equation (HRmax = 220 − age), forest walking corresponded to vigorous-intensity aerobic exercise (70–85% HRmax), whereas track walking corresponded to moderate-intensity exercise (60–70% HRmax). However, the Fox and Haskell equation is an empirical formula that may have an error margin of ±10–12 bpm [42]. As this study was a pilot investigation in the field of public health rather than a clinical trial, it adopted an exploratory approach considering the general health conditions of middle-aged women. Future research should enhance the accuracy of exercise intensity classification by directly measuring individual physiological responses through graded treadmill testing. It is also possible that the slope of the forest trail itself influenced exercise intensity. This study was designed as a self-controlled experiment in which the same participants walked in both environments, minimizing inter-individual differences and allowing direct comparison of environmental effects. However, due to limited statistical power, it should be regarded as a pilot study. For broader generalization, future studies should include a participant control group, standardize exercise intensity using Borg’s RPE scale, and examine how different types of outdoor spaces classified by the presence of greenery and topographical slope affect human health.

Middle-aged women tend to respond sensitively to environmental stimuli due to hormonal and emotional changes associated with the menopausal transition. The finding that the forest environment positively affected emotional stability and cognitive activation may be closely related to these physiological and psychological characteristics. Urban green spaces with high accessibility can function as everyday health resources that promote psychological recovery and stress reduction in middle-aged women. The results of this study provide empirical evidence supporting the need for nature-based urban health promotion strategies that account for age and gender characteristics.

Furthermore, this study is significant in that EEG and heart rate were simultaneously measured in a real urban outdoor setting, allowing for an objective analysis of psychological and physiological responses according to environmental characteristics. However, since the same 30 participants were used and only short-term responses were measured, the statistical power was limited, and long-term effects could not be verified.

In addition, since the study participants were limited to healthy, urban-dwelling middle-aged women, there are limitations regarding the clarity of participant input levels. Future research should analyze how participant characteristics—such as age, gender, occupation type, retirement status, health condition, meditation and physical activity habits, frequency of forest walks, and resilience—correlate with the outcomes. Due to the nature of outdoor experiments, external factors such as temperature, lighting, and noise could not be fully controlled. Future studies should conduct comprehensive evaluations incorporating GIS-based spatial indicators (e.g., green coverage, vegetation density, noise, and temperature) together with additional physiological measures such as heart rate variability (HRV) and oxygen saturation (SpO_2_).

Consequently, this study demonstrates that nature-friendly walking environments, such as forest trails, play a positive role in promoting mental stability and physical health among urban residents. Therefore, urban planning and public health policies should move beyond simply expanding green space areas and consider qualitative design elements such as accessibility, topographical diversity, and multisensory experiences.

Such environments can encourage continuous physical activity among middle-aged populations and contribute to reducing health disparities within urban areas.

## 4. Conclusions

This study analyzed the effects of different walking environments within urban areas on the mental and physical health of middle-aged women. The results showed that forest walking led to significant increases in alpha, beta, and gamma wave activity compared with track walking, indicating greater effects on emotional stability, attention, and cognitive activation. Heart rate was also higher during forest walking, suggesting that it induced a stronger aerobic exercise effect. These findings empirically demonstrate that the qualitative characteristics of walking environments influence the degree and efficiency of both physical and psychological health benefits. In addition, the gentle slope and natural setting of the forest trail contributed to greater exercise intensity and promoted physical health improvement compared with track walking. These results suggest that easily accessible natural environments within cities can substantially contribute to health promotion among middle-aged women, highlighting the importance of securing and designing nature-based exercise spaces as essential elements of urban planning and public health policy. This study was a short-term experimental study conducted with middle-aged women (aged 40–59 years) living in urban areas, and the limited sample size and participant scope constrain the generalizability of the results.

Nevertheless, it provides preliminary evidence supporting the hypothesis that nature-based physical activity in urban settings can enhance both mental and physical health. Future studies should control environmental variables and integrate various physiological indicators with GIS-based spatial metrics to identify objective correlations between environmental quality and health outcomes.

Moreover, it is necessary to examine the differences between subjective perception and physiological responses through self-reported surveys conducted before and after physical activity, which may help clarify the relationship between self-reported measures and objective physiological data such as EEG, heart rate, and HRV. Future studies should include participants of various ages and genders and conduct long-term or repeated experiments to overcome the short-term, single-season limitation and control environmental factors. This will help verify the sustainability of nature-based physical activities and their broader social effects.

In conclusion, future research should adopt an integrative approach to analyze the relationship between environmental characteristics and physiological responses to provide design directions for urban environments that promote public health.

Furthermore, the creation and preservation of accessible, nature-friendly walking environments (e.g., forest trails) should be recognized as key elements of urban infrastructure and public health policy, serving as foundations for promoting daily physical activity and psychological recovery among urban residents.

## Figures and Tables

**Figure 1 healthcare-13-02876-f001:**
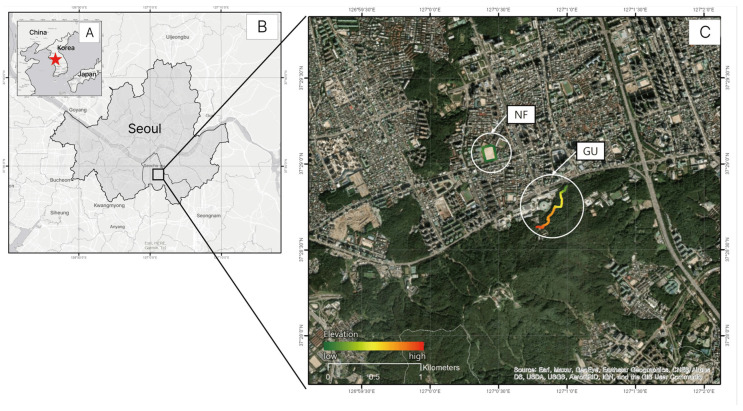
Study area: (**A**) Location of Korea (☆ are location of Seoul Metropolitan City in South Korea), (**B**) Map of Seoul Metropolitan City, (**C**) Seoul High School Playground (NF) located in Seocho-gu, Seoul, and Umyeonsan Mountain barrier-free forest road (GU) located in Korea.

**Figure 2 healthcare-13-02876-f002:**
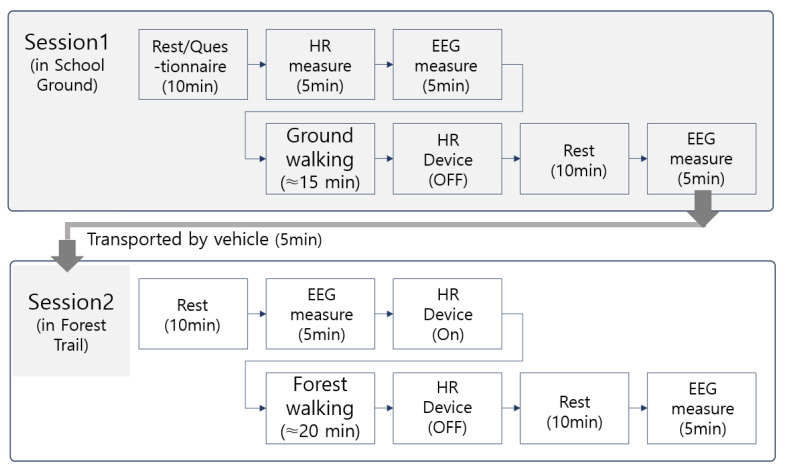
Experiment process.

**Figure 3 healthcare-13-02876-f003:**
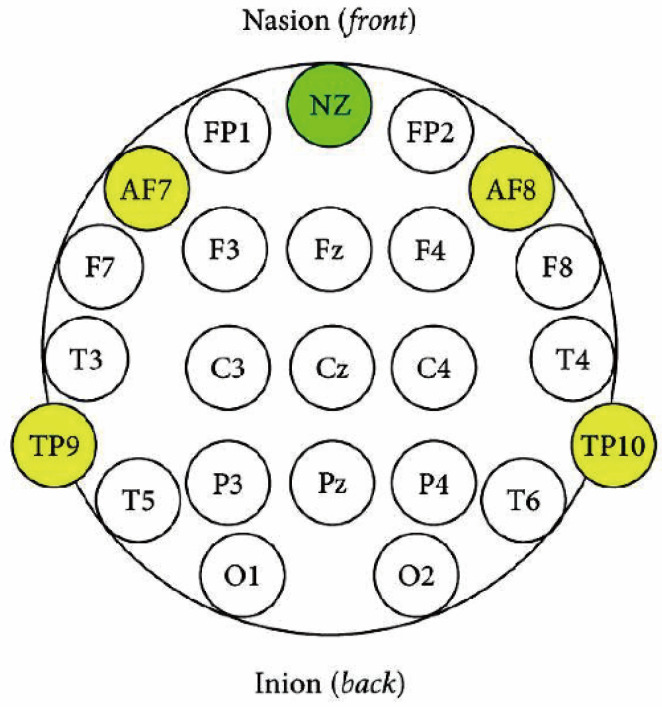
Electrode location for EEG: International 10–20 system.

**Figure 4 healthcare-13-02876-f004:**
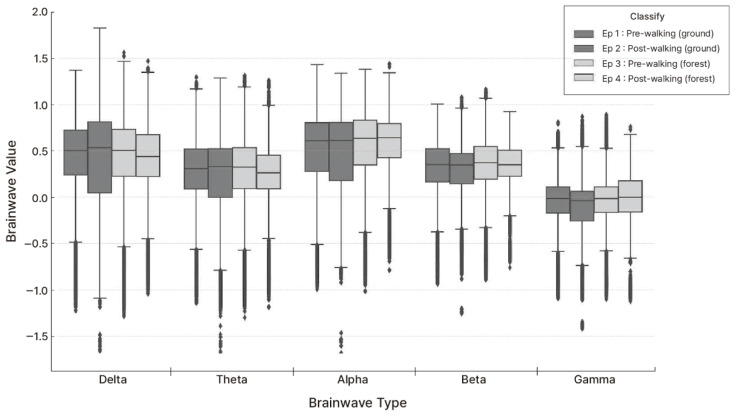
EEG Comparison of 4 Episodes: Visualization table comparing the distributions of five types of brain waves across episodes. The box plots represent the distribution of normalized EEG values for five frequency bands (Delta, Theta, Alpha, Beta, and Gamma). Each box indicates the interquartile range (IQR), with the median shown as a horizontal line. Whiskers represent data within 1.5 × IQR, and dots indicate outliers comparing the distributions of five types of brain waves across episodes.

**Figure 5 healthcare-13-02876-f005:**
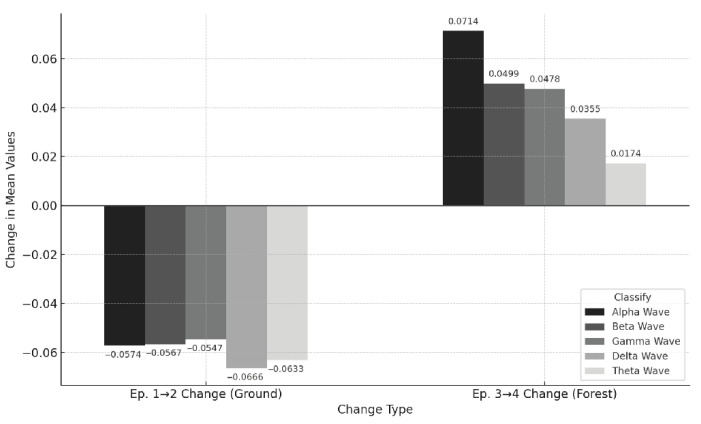
Comparison of EEG Mean Changes by 5 EEG Types: Episode 1 → 2 (pre- and post-walking on the ground) and Episode 3 → 4 (Pre- and post-walking on the ground). Positive values indicate increases after walking; negative values indicate decreases. In the ground, all bands decreased (≈−0.05 to −0.07), with relatively large drops in Delta and Theta. In the forest, all bands increased, with the largest change in Alpha (+0.0714), followed by Beta, Gamma, Delta, and Theta.

**Figure 6 healthcare-13-02876-f006:**
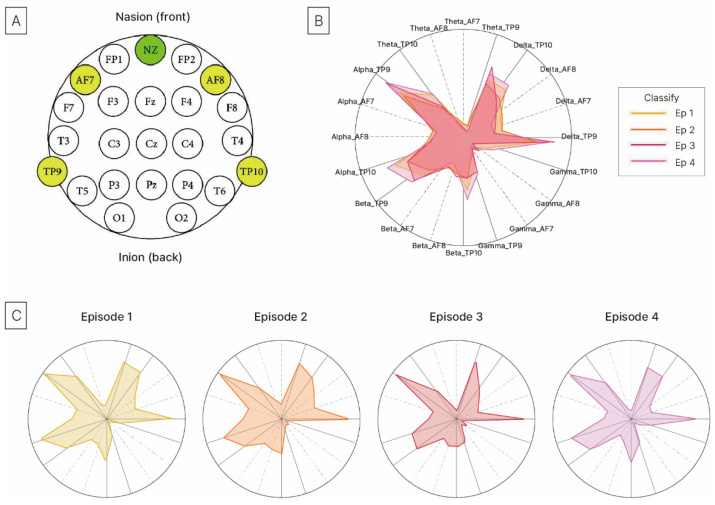
The radar chart: (**A**) Electrode Location for EEG (A green circle represents the front reference point of the head when placing EEG as Nasion (NZ), and the yellow circles indicate the electrode sites (TP9, AF7, AF8, TP10)); (**B**) EEG Intensity Location Episodes (Overlapping Radar Chart); (**C**) EEG Intensity by Electrode Location (Radar Chart by Episodes).

**Figure 7 healthcare-13-02876-f007:**
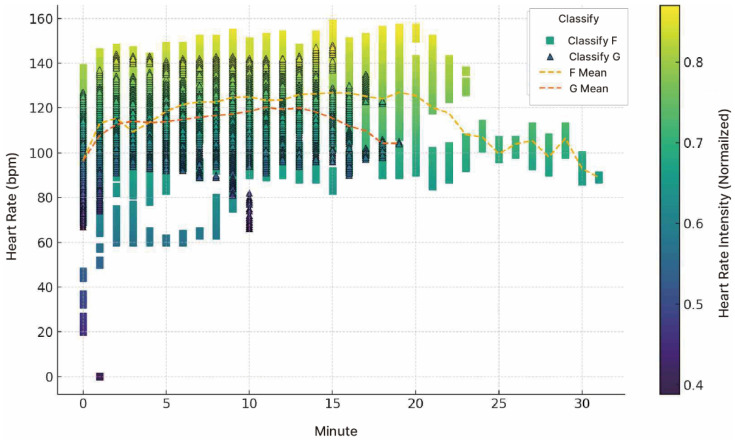
Heart Rate Time Series Analysis: Heart rate (bpm) changed over time while covering the same distance under the forest trail (F) and track (G) walking conditions. The color gradient represents normalized heart rate intensity levels (0.4–0.8), with values closer to 0.8 indicating higher intensity. Data points are displayed as squares for the forest trail (F) and triangles for the track (G), and the dashed lines indicate mean heart rate trends.

**Table 1 healthcare-13-02876-t001:** Study site type.

Study Site		Green	
		Green	Null
Slope	Uphill	GU	NU
	Flat	GF	NF

**Table 2 healthcare-13-02876-t002:** General characteristics of participants.

Category	Age	Physical Activity Status (Days/Week)					Mental Health Status (Days/2 Weeks)			
		Frequency of Physical Activity ≥ 10 min	Moderate-Intensity Exercise	Vigorous-Intensity Exercise	Resistance Training	Flexibility Training	Perceived Stress Level	Perceived Chest Tightness	Headache Severity	Anger Level
Mean	51.2	5.2	2.1	1.7	0.37	1.9	3.3	1.3	1.27	2
SD	3.20	1.97	2.15	2.20	0.81	1.63	2.07	1.71	1.60	1.95

**Table 3 healthcare-13-02876-t003:** Comparison table of EEG results by measurement location.

Site	Episode 1–2 (Ground)	Episode 3–4 (Forest Trail)	Physiological Impact
TP9	Delta and Alpha waves decreased or showed no change	Alpha and Theta waves increased	Sensitive to emotional responses
			Increased responsiveness to forest trails
AF7	Increased responsiveness to forest trails	Increase in Alpha, Theta, Gamma waves	Enhanced responsiveness to emotional and auditory stimuli
AF8	Slight decrease in Alpha, Theta waves	Increase in Alpha, Theta waves	Promotes attention recovery and relaxation
TP10	Slight decrease in Alpha, Theta waves	Increase in Alpha, Theta waves	Promotes attention recovery and emotional stability responses

**Table 4 healthcare-13-02876-t004:** Results of HRR analysis (One-way ANOVA).

Category	Sample	Mean HR (bpm)	SD	Var
Before (B)	30	73.78	7.45	55.44
Ground (G)	30	114.78	9.68	93.72
Forest trail (F)	30	120.69	11.12	123.64

**Table 5 healthcare-13-02876-t005:** Results of the exercise intensity zone analysis.

Category	Mean HR (bpm)	% of HRmax	Exercise Intensity
Before (B)	73.78	43.4	Resting Level (<50%)
Ground (G)	114.78	67.5	Moderate Intensity (60~70%)
Forest trail (F)	120.69	71.0	High-Intensity (70~85%)

## Data Availability

The data are not publicly available due to privacy reasons.

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
