# Peer review of "Effects of Forest Trail and Ground Walking on Mental and Physical Health Promotion in Middle-Aged Women Living in Urban Areas"

_healthcare, 2025, doi:10.3390/healthcare13222876_

Round 1
Reviewer 1 Report
Comments and Suggestions for Authors
The manuscript provides interesting information, but some facts are described insufficiently or not at all.
especially in the methodology section:
-It is not explained who examined the patients? and who set up the examination equipment? (physiotherapist? doctor? research assistant?)
-It is not clear from the text how the walk took place - was it a group walk (all at once? or how big a group?) or individually. If individually, how was it taken into account that each woman could walk in different weather, which could have affected the results. If in group, was the group accompanied by therapist or researcher?
-How was the randomization process carried out?
-How were the women recruited? Were they volunteers approached through a leaflet? Or, for example, on the recommendation of a doctor? It is necessary to specify (again, it can affect the results). What were inclusion and exclusion criteria?
-What are the characteristics of the sample related to this point? A table describing the demographics of the participants should be included in the manuscript. Did they have any comorbidities? Did they take any medications?
-was there any drop out among the study participants?
-in the introduction section, it should be explained in more detail why the authors think that walking will have an effect on the EEG activity of the brain? and then in the discussion, comment on whether the changes are clinically significant and relevant?
-in conclusion, I lack the implication for clinical practice
-in figure 3 and 4 I am missing labels explaining what each box plot indicates.
Author Response
General Comments: The manuscript provides interesting information, but some facts are described insufficiently or not at all.
Response: We appreciate the valuable review and thoughtful feedback. We have addressed all of the reviewer’s points and clarified the procedural details of the Methods section. The specific revisions are described in the following items.
Comments 1: It is not explained who examined the patients? and who set up the examination equipment? (physiotherapist? doctor? research assistant?)
Response 1: We appreciate the insightful comment. All measurement procedures in this study were jointly performed by three research assistants skilled in the use of physiological measurement devices for physical-activity assessment and the principal investigator (PI). The research assistants, who had received prior training, were responsible for EEG setup and for measuring heart rate (HR) the entire procedure was supervised by the PI.
(Revised section: 2.3. Methods (2.3.1 Methods of measurement), Lines 156–162)
Comments 2: It is not clear from the text how the walk took place - was it a group walk (all at once? or how big a group?) or individually. If individually, how was it taken into account that each woman could walk in different weather, which could have affected the results. If in group, was the group accompanied by therapist or researcher?
Response 2: We sincerely appreciate the reviewer’s valuable comment.
This study was conducted using an individual walking approach. Each participant performed the walking sessions under similar weather conditions (clear sky, wind speed below 3 m/s) and within the same temperature range (18–20°C) between 10 a.m. and 5 p.m. on the same week to minimize the influence of weather variability. In addition, all experiments were conducted under the supervision of the researcher, who accompanied the participants on-site to ensure safety and accurate measurement.
(Revised section: 2.3. Methods (2.3.1 Methods of measurement), Lines 169–174)
Comments 3: How was the randomization process carried out?
Response 3: We sincerely appreciate the reviewer’s insightful comment.
Random assignment was performed using a computer-based random number generator, and all participants completed the walking sessions in the order of ground walking followed by forest walking.
(Revised section: 2.3. Methods (2.3.1 Methods of measurement), Lines 163–168)
Comments 4: How were the women recruited? Were they volunteers approached through a leaflet? Or, for example, on the recommendation of a doctor? It is necessary to specify (again, it can affect the results). What were inclusion and exclusion criteria?
Response 4: Participant recruitment was conducted through an open call using flyers, based on a non-probability sampling method (conditional recruitment). All participants were voluntarily recruited middle-aged women. The inclusion criterion specified on the flyer was women aged 40–59 years, with no specific restrictions regarding medical conditions. However, individuals who (1) had physical disabilities or (2) had difficulty walking more than 2 km were excluded.
(Revised section: 2.2. Participants, Lines 127–131)
Comments 5: What are the characteristics of the sample related to this point? A table describing the demographics of the participants should be included in the manuscript. Did they have any comorbidities? Did they take any medications?
Response 5: We have added a summary of the participants’ demographic characteristics.
Information on mean age, presence of physical and mental illness, and walking activity was included; however, height and weight data, which could be considered sensitive, were not collected.
The average age of the participants was 51.2 years, and all were healthy women without any physical or mental disorders and were not taking medication.
According to the brief survey results, all participants reported walking for more than one hour on an average of 5.2 days per week, whereas the frequency of moderate- or vigorous-intensity exercise was relatively low, averaging about two days per week.
(Revised section: 2.2. Participants, Lines 138–142)
Comments 6: was there any drop out among the study participants?
Response 6: There were no dropouts during the study period. All 30 participants completed the walking experiments in both environments.
(Revised section: 2.2. Participants, Lines 142–143)
Comments 7: in the introduction section, it should be explained in more detail why the authors think that walking will have an effect on the EEG activity of the brain?
Response 7: We sincerely appreciate the reviewer’s highly important comment.
In the Introduction section, we have supplemented the study objective to clarify that this research was developed based on previous findings indicating that walking in natural environments induces prefrontal activation and increases alpha wave activity, thereby contributing to mental relaxation and cognitive enhancement (Berman et al., 2012; Ulrich et al., 1991). Accordingly, the revised manuscript emphasizes the aim of examining changes in EEG according to different walking environments.
(Revised section: 1. Introduction, Lines 76–80)
Comments 8: and then in the discussion, comment on whether the changes are clinically significant and relevant? in conclusion, I lack the implication for clinical practice
Response 8: We appreciate the reviewer’s valuable comment. In addition, we have added a statement indicating that the observed EEG changes can be supported by previous studies showing their significant relationship with psychological relaxation and attention recovery. However, this study was conducted to verify, through quantitative data, how the health-promoting effects of walking—an easily accessible physical activity in daily life—are related to the activity environment. Therefore, we would like to ask for understanding that this study has a preliminary nature with public health and environmental significance rather than clinical meaning.
These contents were added and explained in Section 3.3 Discussion, and we also strengthened the practical implications by specifying that the results of this study can be utilized in designing mental health promotion programs for middle-aged women and in establishing policies for creating therapeutic walking trails in urban areas.
(Revised section: 3.3. discussion, Lines 411–424)
Comments 9: in figure 3 and 4 I am missing labels explaining what each box plot indicates.
Response 9: We have clearly indicated all waveform types (Delta, Theta, Alpha, Beta, and Gamma) and environmental conditions (Ground vs. Forest) on the axes and legends of Figures 4 and 5. In addition, the meaning of each graph has been described in detail both in the main text and in the corresponding figure captions.
(Revised section: 3.1. EEG Results(3.1.1. Pre- and post-walking EEG changes) Lines 262–266, 281–285)
Summary of the answer
• Specified the measurement personnel and procedures in detail
• Clarified walking conditions, including group composition, duration, and weather control
• Added a description of the random assignment process
• Elaborated on participant recruitment procedures and inclusion/exclusion criteria
• Added a table presenting participant characteristics (Table 2)
• Strengthened the theoretical and clinical foundations in the Introduction and Discussion
• Revised and supplemented figure labels and legends for clarity

Reviewer 2 Report
Comments and Suggestions for Authors
The work is interesting in general, but there are a number of shortcomings:
The summary mentions Covid – it is unclear why this is necessary since there is not a word about Covid in the text!
Lines 58 -60 for what the description of the methodology is presented in the introduction of the work.
Lines 60 - 65 the purpose of the study is not specific, it just describes the research methodology.
In addition, there is no information about the effect of walking on EEG in the introduction
Lines 69 -77 it is unclear why this text is needed and there is no description of the methodology!
In addition, the walking paths used in the method have different levels of inclination!!\
Lines 103 - the description of the group is incorrect! There is no understanding of what kind of women they are! Their age, weight, height, and BMI. Have they been sick with Covid? How long have they been sick, when did they get sick, and what is the extent of their lung damage?
Lines 111 - it is unclear who provided the ethical approval for the study.
Lines 120 - 123 - it is unclear why this information is needed, and where is the description of the heart rate monitor?
Why is the slope described approximately??
Lines 127 - 134 - it is unclear why this text is included.
The use of EEG - who applied the electrodes? How were the underlying tissues prepared? Was the same researcher responsible for applying the electrodes?
Lines 156 - it is unclear how the heart rate monitor was used. Where was it placed?
Lines 158 - 163 - describe what descriptive statistics are presented? There is no P level!!
The methodology should be rewritten and structured! There is no accurate description of the women! THERE IS NOT A SINGLE WORD ABOUT WHETHER THEY HAVE COVID!!!!
It is unclear why the text is presented on Lines 178 - 180.
Why are there no questionnaire survey data?
Lines 259 - 261 - what do the authors attribute the changes in HR to? maybe it's just the influence of the slope?!!!
In general, it is necessary to significantly improve the article, while the data obtained during the study are interesting and can be submitted for publication.
Author Response
Comments 1: The summary mentions Covid – it is unclear why this is necessary since there is not a word about Covid in the text!
Response 1: Thank you for your comment. The mention of “the importance of physical activity after the COVID-19 pandemic” in the abstract was intended to provide social context for the study. However, to maintain consistency with the main text, all references to COVID-19 have been removed.
(Revised section: Abstract, Lines 10 deleted)
Comments 2: Lines 58–60: for what the description of the methodology is presented in the introduction of the work.
Response 2: We appreciate the reviewer’s comment. The sentence related to the research method, which was included in the Introduction (lines 57–59), was considered to be in an inappropriate position and has therefore been deleted.
(Revised section: 1. Introduction, Line 3 deleted)
Comments 3: Lines 60–65: the purpose of the study is not specific, it just describes the research methodology.
Response 3: To clarify the purpose of the study, the statement has been revised as follows:
“The purpose of this study is to compare changes in EEG and heart rate according to walking environments (ground vs. forest) among middle-aged women living in urban areas, in order to identify the effects of natural environments on mental and physical health.”
(Revised section: 1.Introduction, Lines 85–87)
Comments 4: In addition, there is no information about the effect of walking on EEG in the introduction.
Response 4: In response to the reviewer’s comment, we have added supporting evidence from previous studies.
“Previous studies have reported that walking exercise increases alpha and beta wave activity in the prefrontal cortex, which positively affects attention and cognitive function (e.g., Thompson et al., 2011). Accordingly, this study aimed to experimentally examine whether similar results could be observed depending on the walking environment.”
(Revised section: 1. Introduction, Lines 76–84 added)
Comments 5: Lines 69–77: it is unclear why this text is needed and there is no description of the methodology!
Response 5: This paragraph described the rationale for selecting easily accessible outdoor urban environments instead of indoor spaces to verify the health-promoting effects among middle-aged women living in cities. Unnecessary content has been deleted and refined, and the detailed explanation of the research design has been integrated into the Methods section.
(Revised section: 2.1. Study area, Lines 91–94)
Comments 6: In addition, the walking paths used in the method have different levels of inclination!
Response 6: Thank you for your comment. We have added a clear explanation regarding the slope characteristics of the walking routes. The forest trail was established on a mountain located within the city; therefore, due to its natural topographical features, the slope may vary. This point has been clarified in the manuscript. In addition, since this study employed a within-subject design in which the same participants walked in both environments (a green area with slopes and a flat area without greenery), we have mentioned that the study also aimed to examine the effects associated with the presence or absence of slopes.
(Revised section: 2.1. Study area, Lines 105–113)
Comments 7: Lines 103: The description of the group is incorrect! There is no understanding of what kind of women they are! Their age, weight, height, and BMI. Have they been sick with Covid? How long have they been sick, when did they get sick, and what is the extent of their lung damage?
Response 7: We have added a table presenting the demographic and health characteristics of the participants. The mention of COVID-19 was intended only to provide social context; therefore, information regarding participants’ COVID-19 infection status was not collected, as it was not directly relevant to the purpose of this study. Participants were recruited among middle-aged women who were physically capable of walking at the time of recruitment, and we specified their mean age as well as the absence of any physical or mental disorders. However, height and weight data were not collected because they were considered unrelated to the main focus of this study and potentially sensitive information.
(Revised section: 2.2. Participants, Lines 147)
Comments 8: Lines 111: it is unclear who provided the ethical approval for the study.
Response 8: We have specified the institution that approved the research ethics.
“This study was approved by the Institutional Review Board of Korea University (KUIRB-2025-000501), and all participants provided written informed consent and consent for the collection, use, and provision of personal information prior to participation.”
(Revised section: 2.2. Participants, Lines 144–146)
Comments 9: Lines 120–123 - it is unclear why this information is needed,
Response 9: The content in lines 119–122 was originally included to explain the rationale for selecting EEG and HR as indicators to quantitatively measure the effects of walking in natural environments on mental and physical health. However, to improve the clarity and logical flow of the manuscript, this explanation has been moved to the Introduction section, where the context of the research design is more clearly presented.
(Revised section: 1. Introduction, Lines 54–68)
Comments 10: and where is the description of the heart rate monitor?
Response 10: We have clarified the description of the equipment used in the study.
“Heart rate was measured using a Polar H10 heart rate monitor chest strap (Manufacturer: Polar Electro, Finland) worn on the chest. All participants attached the device under the guidance of a trained research assistant. Before measurement, the sensor position was secured, and the skin contact area was disinfected with an alcohol swab. Measurements were conducted under the supervision of the principal investigator.”
(Revised section: 2.3. Methods (2.3.3 Physical health measurement), Lines 206–210)
Comments 11: Why is the slope described approximately?
Response 11: In selecting the study site, the forest trail was chosen for its easy accessibility to middle-aged women living in urban areas and for being a gently sloped path that anyone can walk. We have clarified that, due to the natural topographical characteristics, the slope of the trail is gentle but not uniform, and this variation was expressed as a range of slope values in the manuscript.
(Revised section: 2.1. Study area, Lines 95–99)
Comments 12: Lines 127 - 134 - it is unclear why this text is included.
Response 12: The sentence was originally written to explain the rationale for selecting EEG as an indicator for measuring mental health. However, since this rationale was already sufficiently described in the Introduction, it was summarized to avoid redundancy.
(Revised section: 2.3. Methods (2.3.3. Physical health measurement), Lines 180–183)
Comments 13: The use of EEG - who applied the electrodes? How were the underlying tissues prepared? Was the same researcher responsible for applying the electrodes?
Response 13: Thank you for your comment. We have clearly described the measurement procedure as follows:
“EEG electrode attachment was performed uniformly for all participants by a trained research assistant. Participants were seated comfortably in a chair with a backrest, kept their eyes closed, and remained still during the measurement. EEG measurements were conducted in a quiet environment free from external noise and disturbance by others.”
(Revised section: 2.3. Methods (2.3.2. Mental health measurement), Lines 192–197)
However, we have acknowledged as a study limitation that, because the measurements were conducted in an outdoor environment, the setting could not be completely isolated from various external influences.
(Revised section: 3.3. discussion, Lines 433–439)
Comments 14: Lines 156 - it is unclear how the heart rate monitor was used. Where was it placed?
Response 14: We have clarified the chest attachment position and data collection procedure as follows:
“For heart rate measurement, the sensor was attached to the center of the chest, and data were collected in real time. To assess the resting heart rate before exercise, participants sat in a chair with a backrest while the measurement was taken. Subsequently, each participant wore the heart rate monitor and performed walking sessions on both the track and the forest trail. Heart rate data were continuously recorded in real time from the beginning to the end of walking through a mobile application.”
(Revised section: 2.3. Methods (2.3.3 Physical health measurement), Lines 211–221)
Comments 15: Lines 158–163 - describe what descriptive statistics are presented? There is no P level!
Response 15: We have clearly revised the description of the statistical analysis procedures as follows:
“All statistical analyses were performed using the Data Analysis ToolPak in Microsoft Excel. First, independent sample t-tests were conducted to compare the differences before and after walking within each environment (track and forest trail). Subsequently, a one-way analysis of variance (ANOVA) was performed to examine the differences among the four episodes (before and after walking on the track and before and after walking on the forest trail). For each variable, the mean (M), standard deviation (SD), and variance (s²) were calculated, and the significance level was set at p < 0.05.”
(Revised section: 2.4. Analysis, Lines 223–229)
Comments 16: The methodology should be rewritten and structured!
Response 16: We sincerely appreciate the reviewer’s careful review. The content regarding the experimental procedures and methods has been clearly revised. The methodological section has been supplemented, and the experimental process has been presented in a structured format as Figure 2.
(Revised section: 2.3. Methods (2.3.1 Methods of measurement), Lines 177)
Comments 17: There is no accurate description of the women! THERE IS NOT A SINGLE WORD ABOUT WHETHER THEY HAVE COVID!!!!
Response 17: Information regarding participant recruitment, demographics, and health status has been added to the “Participants” section. According to the preliminary survey results, the participants had a mean age of 51.2 years, and all were healthy women without any physical or mental disorders and were not taking medication. On average, participants engaged in walking activities for more than one hour per day on approximately 5.2 days per week, while the frequency of moderate- to vigorous-intensity exercise was about twice per week.
The mention of COVID-19 was intended only to provide social and temporal context; therefore, information regarding infection status or related diseases was not collected, as it was not relevant to the purpose of this study. We would like to clarify that this paper focuses on the relationship between everyday environments, health, and well-being, rather than having a clinical or medical objective.
(Revised section: 2.2. Participants, Lines 137–142)
Comments 18: It is unclear why the text is presented on Lines 178 – 180.
Response 18: Thank you for pointing out this serious mistake. The content was entirely unrelated to this paper and has been deleted.
(Revised section: 3.1. EEG Results(3.1.1. Pre- and post-walking EEG changes) Lines deleted)
Comments 19: Why are there no questionnaire survey data?
Response 19: The purpose of this study was to quantitatively identify the effects of physical and mental health promotion using objective physiological indicators. Subjective perception surveys after the experiment were not included in this study. However, a pre-experimental questionnaire was conducted prior to the experiment. This questionnaire included items on demographic characteristics such as age and residential area, as well as existing health status, habitual physical activity level, and stress level. These details have been added to the participant information section.
(Revised section: (2.2. Participants, Lines 137–143),
In addition, we have added a statement suggesting that future studies should include self-reported questionnaires administered before and after physical activity to examine the relationship and differences between subjective perceptions and physiological responses.
(Revised section: 4. conclusion, Lines 476–479)
Comments 20: Lines 259–261 - what do the authors attribute the changes in HR to? maybe it's just the influence of the slope?!
Response 20: You are correct. The increase in exercise intensity due to the slope of the forest trail is likely to have influenced the rise in heart rate. We have clearly stated this point in the manuscript as follows:
“The increase in heart rate observed in the forest environment may be attributed to the higher exercise intensity caused by the slope (average 5%), which could have affected the physiological responses.”
(Revised section: 3.2. Heart rate results (3.2.3. Time-series analysis of heart rate changes), Lines 371–376)
Final Comment: In general, it is necessary to significantly improve the article, while the data obtained during the study are interesting and can be submitted for publication.
Summary of the answer: We sincerely appreciate the reviewer’s valuable comments. The manuscript has been thoroughly revised to reflect the detailed suggestions provided.
The following major revisions have been made:
• Removal of unnecessary references to COVID-19
• Clarification of participant information
• Specification of measurement instruments and procedures
• Clear presentation of statistical methods and p-values
• Supplementation of figure and table descriptions

Reviewer 3 Report
Comments and Suggestions for Authors
General Comments: The current manuscript seeks to examine the role of two different types of physical environments on walking activity and the outcomes of changes supporting mental and physical health among middle-aged Korean women (40 -59 yrs). The authors provide sufficient detail regarding study design and methods of measuring mental and physical health outcomes, through mobile EEG measures and heart rate and blood pressure responses, respectively. Study subjects were 30 middle-aged women who were exposed to both environmental settings while walking, a school yard or grounds and a forest trail. Hence, each subject served as their own control. All subjects completed their monitored walks under both experimental conditions - grounds and forest trail. Results were detailed regarding EEG, Heart rate, and BP findings showing that the EEG patterns demonstrated improved 'stress' control and other signs consistent with improved mental health during the forest trail walk vs. the 'grounds' walk. Heart rate and BP measures were consistently greater during the forest walk, indicating that the intensity of the walking was increased from a moderate level for the grounds to a more vigorous level for the forest walk. Conclusions/discussion were well presented, however, limitations associated with this study were not adequately discussed.
Specific comments:
The sample size of 30 individuals serving as their own control is a weaker study design and may lack enough power, statistically. Hence, the study should be considered a 'pilot study at best.
The use of the formula 220-age to predict maximal heart rate is a poor measure, since maximal heart rate varies considerably across ages and genders. The formula was developed in Sweden using male physical education students. So, a better approach is to actually measure max heart rate among subjects through exercise testing by treadmill. Also maximum heart rate is dramatically affected by certain medications used to treat hypertension and Ischemic Heart Disease. What chronic conditions, if any, were prevalent among your subjects? Reporting of their health status is rather important for such a study.
Your conclusions regarding the intensity of the walking being higher on the forest trail, need further interpretation. Did subjects volitionally increase their efforts during the forest walk or was this solely due to the intermittent 5% incline? Your conclusion that the more vigorous intensity was responsible for the improved EEG findings cannot be substantiated unless you control for intensity - i.e., the walking effort should be exactly the same for each session with only the environment being different. This can be controlled for by using ratings of perceived exertion (RPE) where the RPE is kept the same for both environmental conditions.
Author Response
General Comments: The current manuscript seeks to examine the role of two different types of physical environments on walking activity and the outcomes of changes supporting mental and physical health among middle-aged Korean women (40 -59 yrs). The authors provide sufficient detail regarding study design and methods of measuring mental and physical health outcomes, through mobile EEG measures and heart rate and blood pressure responses, respectively. Study subjects were 30 middle-aged women who were exposed to both environmental settings while walking, a school yard or grounds and a forest trail. Hence, each subject served as their own control. All subjects completed their monitored walks under both experimental conditions - grounds and forest trail. Results were detailed regarding EEG, Heart rate, and BP findings showing that the EEG patterns demonstrated improved 'stress' control and other signs consistent with improved mental health during the forest trail walk vs. the 'grounds' walk. Heart rate and BP measures were consistently greater during the forest walk, indicating that the intensity of the walking was increased from a moderate level for the grounds to a more vigorous level for the forest walk. Conclusions/discussion were well presented, however, limitations associated with this study were not adequately discussed.
Response: We sincerely appreciate the reviewer’s valuable comments. The study limitations have been revised and supplemented in the Conclusion section of the updated manuscript. We have specified that, although the study employed a within-subject design in which the same participants performed walking tasks in both environments, the limited sample size characterizes this research as a pilot study. In addition, we have noted that the generalizability of the findings is limited.
(Revised section: 4. Conclusion, Lines 479–482)
Comments 1: The sample size of 30 individuals serving as their own control is a weaker study design and may lack enough power, statistically. Hence, the study should be considered a 'pilot study‘ at best.
Response 1: We agree with the reviewer’s comment. In the revised manuscript, the Conclusion section clearly states that this study is a preliminary exploratory study (pilot study) and that caution should be exercised when generalizing the results due to the limited sample size. In addition, regarding the use of each participant as their own control, we have noted that future studies should consider standardizing exercise intensity across environments by employing Borg’s Rating of Perceived Exertion (RPE) scale.
(Revised section: 3.3. discussion, Lines 417–424)
Comments 2: The use of the formula 220-age to predict maximal heart rate is a poor measure, since maximal heart rate varies considerably across ages and genders. The formula was developed in Sweden using male physical education students. So, a better approach is to actually measure max heart rate among subjects through exercise testing by treadmill. Also maximum heart rate is dramatically affected by certain medications used to treat hypertension and Ischemic Heart Disease. What chronic conditions, if any, were prevalent among your subjects? Reporting of their health status is rather important for such a study.
Response 2: We sincerely appreciate the reviewer’s valuable comments. In this study, the Fox and Haskell equation (HRmax = 220 − age) was applied, as it is one of the most widely used empirical formulas for prescribing exercise intensity (Robergs & Roberts, 2002). As reported in several studies, this equation is an empirical estimation derived from simplified population data and may include a certain degree of error (Shookster et al., 2020). However, this study was conducted as a public health-oriented investigation focusing on daily physical activity and environmental factors rather than a clinical trial. It is therefore positioned as a preliminary study that considers the general health status of middle-aged women.
All participants were healthy middle-aged women without any specific diseases, as confirmed through the pre-survey. Thus, we applied the Fox and Haskell equation, which has been reported to have relatively low risk of over- or underestimation and to be suitable for general populations (Shookster et al., 2020). We would like to ask for understanding regarding this methodological choice.
We have acknowledged the limitations of the “220 − age” formula and added a statement in the Discussion section suggesting that future studies should utilize data from graded exercise tests (GXT) obtained through treadmill-based assessments.
(Revised section: 3.3. discussion, Lines 411–419)
Comments 3: Your conclusions regarding the intensity of the walking being higher on the forest trail, need further interpretation. Did subjects volitionally increase their efforts during the forest walk or was this solely due to the intermittent 5% incline? Your conclusion that the more vigorous intensity was responsible for the improved EEG findings cannot be substantiated unless you control for intensity - i.e., the walking effort should be exactly the same for each session with only the environment being different. This can be controlled for by using ratings of perceived exertion (RPE) where the RPE is kept the same for both environmental conditions.
Response 3: We sincerely appreciate the reviewer’s insightful and important comment. In this study, to observe physiological responses according to environmental differences in physical activity, no intensity control was implemented; participants were simply instructed to walk at a self-selected pace that was slightly faster than their usual walking speed. However, reflecting the reviewer’s suggestion, we have added a statement in Section 3.3 (Discussion) acknowledging that the 5% slope and natural terrain characteristics of the forest trail may have influenced exercise intensity. Furthermore, to clarify whether EEG changes were solely attributable to differences in intensity, we have suggested that future studies should employ Borg’s Rating of Perceived Exertion (RPE) scale to control walking effort.
(Revised section: 3.3. discussion, Lines 406–410, 419–424)
Summary of the answer
We sincerely appreciate the reviewer’s detailed and professional advice. We believe that these revisions have greatly contributed to enhancing the clarity and reliability of the manuscript. In accordance with the reviewer’s suggestions, the following revisions have been made:
• Clear description of the study design limitations
• Inclusion of a statement addressing the limitations of the calculation formula used for the results
• Supplementation of the discussion regarding participant characteristics and exercise intensity control

Round 2
Reviewer 2 Report
Comments and Suggestions for Authors
The authors have revised the article, and it is generally clear that the study is promising and relevant. However, there are some concerns:
The conclusion that psychological well-being improves solely based on changes in wave activity may be somewhat controversial, and it would be beneficial to support it with psychological tests, cognitive methodologies, and other evidence. Additionally, more thorough monitoring is necessary to determine the long-term impact of this positive effect.
It is not entirely clear what the "accessibility" of the environment is referring to - in the study, respondents were transported by vehicles, and it is clear that walking in a forest area is very beneficial, but the practical question is how easy it is for city dwellers to access such recreational areas (i.e., is there a forest somewhere that they can go to for a walk, or is there a park within the city limits that they can use for convenience? This aspect is somewhat unclear).
It is not very clear how different the level of input is for the study participants - everyday physical activity (maybe someone only goes to the store. but with heavy bags and far away, and someone does exercises every day and runs on the track), psychological stress (who has a stressful job, and who is already retired, or who practices meditation and can relax faster in principle, and for whom this forest walk is the only release)
In general, the impression of the article is positive, but it looks a little utopian - as if everyone has a forest on a slope within walking distance and the opportunity to walk there regularly
In addition - shortcomings that were also in the first version of the work - no control group, short observation period, insufficiently detailed procedures for calibrating equipment, selection criteria are not strict enough (lack of information about previous experience of physical activity), insufficiently controlled external factors (weather and time of day), absent the information about the participants' usual level of physical activity.
Study limitations: single measurement of indicators, no consideration of seasonal factors, no study of long-term effects, and insufficiently detailed description of participant characteristics!!
Author Response
Comments 1: The conclusion that psychological well-being improves solely based on changes in wave activity may be somewhat controversial, and it would be beneficial to support it with psychological tests, cognitive methodologies, and other evidence. Additionally, more thorough monitoring is necessary to determine the long-term impact of this positive effect.
Response 1: We sincerely appreciate your valuable comment.
We also agree that it is scientifically insufficient to conclude that psychological well-being was improved solely based on EEG changes.
Nevertheless, the results of this study showed tendencies consistent with previous research in two aspects.
First, it has been repeatedly reported in several neurophysiological studies that an increase in alpha (α) wave activity is related to emotional stability and psychological relaxation (Davidson, 2004).
Second, the EEG change pattern observed in this study was consistent with previous findings that an increase in alpha wave activity in the temporal–parietal(TP) regions is closely related to psychological well-being and emotion-regulation functions (Kim et al., 2022).
These findings suggest the possibility that walking in natural environments activates the emotion-regulation–related areas of the brain and contributes to psychological stability.
However, as this study is a pilot study, it has limitations in clarifying causal relationships; therefore, we have added in the limitation section that we plan to conduct an integrated analysis combining subjective and objective indicators by using psychological scales (e.g., POMS, PANAS) and cognitive tasks (e.g., working memory test), and to verify the lasting effects through longitudinal monitoring.
(Revised section: 3.3. Discussion, Lines (427-433)
Comments 2: It is not entirely clear what the "accessibility" of the environment is referring to - in the study, respondents were transported by vehicles, and it is clear that walking in a forest area is very beneficial, but the practical question is how easy it is for city dwellers to access such recreational areas (i.e., is there a forest somewhere that they can go to for a walk, or is there a park within the city limits that they can use for convenience? This aspect is somewhat unclear).
Response 2: We sincerely appreciate the reviewer’s valuable comment.
In this study, the term “accessibility” was used not simply to indicate the convenience of movement through transportation but as a concept encompassing the availability of natural environments that urban residents can easily access within their daily living areas.
The forest trail (GU), which served as the experimental site, is a barrier-free trail located within Umyeonsan Mountain (maximum elevation 293 m) in the central area of Seocho-gu, Seoul.
As an urban mountain area surrounded by residential and commercial districts, nearby residents can easily reach it on foot or by public transportation, and it is in fact widely used as a space for daily walking and health-related activities.
In addition, this forest trail was constructed as a safe walking route with a gentle slope of approximately 5%, making it an urban natural area that can be easily used without difficulty by middle-aged women and mobility-vulnerable groups such as the elderly.
The fact that participants used vehicles on the day of the experiment was a procedural step in the research design to control potential urban environmental factors that could arise during movement between the school ground and the forest trail, and in reality, the site has very high accessibility within the local residents’ daily living area.
(Revised section: 2.1. Study area, Lines (103-110)
Comments 3: It is not very clear how different the level of input is for the study participants - everyday physical activity (maybe someone only goes to the store. but with heavy bags and far away, and someone does exercises every day and runs on the track), psychological stress (who has a stressful job, and who is already retired, or who practices meditation and can relax faster in principle, and for whom this forest walk is the only release)
Response 3: We sincerely appreciate the reviewer’s valuable comment.
As mentioned by the reviewer, differences in participants’ habitual physical activity levels and psychological states are important factors that may influence the research results.
Accordingly, in this study, we confirmed the participants’ basic health conditions and lifestyle habits through a preliminary questionnaire.
All participants were healthy middle-aged women (aged 40–59 years) who had no specific diseases and regularly engaged in daily walking activities.
According to the results presented in Table 2, all participants performed walking for more than one hour on an average of 5.2 days per week,
whereas the frequencies of moderate-intensity exercise (2.1 days) and vigorous-intensity exercise (1.7 days) were relatively low.
This indicates that the participants were a group maintaining a general level of health with a lifestyle centered on light-to-moderate physical activity.
In addition, the self-reported levels of stress (3.3 days), chest tightness (1.3 days), headache (1.27 days), and anger (2.0 days) were based on the number of days these symptoms were felt during the past two weeks, showing overall low levels. Therefore, it can be confirmed that the participants took part in the experiment in a generally stable psychological condition.
(Revised section: 3.3. Discussion, Lines (131-149)
However, there is a limitation in that detailed psychological and lifestyle factors, such as participants’ occupational types and meditation habits, could not be completely controlled.
This point has been specified in the Limitations section, and in future research, we plan to include variables such as occupational group, stress level, and resilience in order to analyze EEG response differences according to participant characteristics in a more precise manner.
(Revised section: 3.3. Discussion, Lines (478-482)
Comments 4: In general, the impression of the article is positive, but it looks a little utopian - as if everyone has a forest on a slope within walking distance and the opportunity to walk there regularly
Response 4: We sincerely appreciate the reviewer’s insightful comment.
As correctly noted, it is indeed difficult for all urban residents to use forest trails or gently sloped walking paths on a daily basis.
However, the aim of this study was not to present such environments as ideal or utopian settings, but to scientifically examine the mental and physical health benefits of forest trails—natural environments with forested and gently sloped terrain that are realistically accessible within the daily living areas of urban residents.
The experimental site, the Umyeonsan forest trail (GU), is not a designated recreational forest or suburban mountain area, but a barrier-free trail developed within a mountain located in the central district of Seocho-gu, Seoul, and thus easily accessible to local residents.
Accordingly, the results of this study should not be interpreted as assuming that all urban residents can routinely access forest environments, but rather as empirical evidence that the utilization of urban forests with natural topography can contribute to citizens’ mental stability and physical health improvement.
(Revised section: 2.1. Study area, Lines (103-110)
Comments 5: In addition - shortcomings that were also in the first version of the work - no control group, short observation period, insufficiently detailed procedures for calibrating equipment, selection criteria are not strict enough (lack of information about previous experience of physical activity), insufficiently controlled external factors (weather and time of day), absent the information about the participants' usual level of physical activity.
Response 5: We sincerely appreciate the reviewer’s valuable comment.
We acknowledge all the points raised as limitations of this study, and we have supplemented or clearly described them in the Limitations section.
1. no control group
As pointed out, this study did not include an independent control group of participants.
However, the study classified outdoor spaces based on the presence of greenery (green) and topographical slope (uphill or flat) (Table 1) to verify the health-promoting effects of physical activity in forest environments with natural terrain.
All participants walked in both environments (school ground and forest trail), minimizing inter-individual physiological differences and enabling a direct comparison of how environmental factors affect human physiological responses. Therefore, in this study, the school track (NF) can be regarded as a comparative condition, that is, a functional control for the forest trail (GU).
(Revised section: 2.1. Study area, Lines 91–100)
Nevertheless, we recognize that the absence of an independent control group limits the generalizability of the findings.
Accordingly, we have revised the Limitations section to state that “this study is a pilot study, and future research should adopt an extended design including an independent control group.”
Furthermore, we specified that future studies will conduct walking experiments across four environmental types (NU, NF, GU, and GF) presented in Table 4 to comprehensively examine how environmental differences influence human mental and physical health.
(Revised section: 3.3. discussion, Lines (453-459)
2. Short observation period
This study has a limitation in that it was conducted as a single (one-time) experiment during the same season, and therefore could not observe long-term changes. This limitation reflects the fact that the present experiment was a preliminary study; in future research, we plan to verify the sustained effects through longitudinal or repeated-measurement studies.
We have added this statement to conclusion sections.
(Revised section: 4. Conclusion Lines (522-526)
3. Equipment calibration
The measurement procedure was conducted in two sessions, first on the school ground and then on the forest trail, and all procedures were carried out by a research team consisting of three trained research assistants and the principal investigator.
The research assistants, who had received prior training in the use of physiological monitoring instruments for physical activity assessment, were responsible for EEG device setup and heart rate measurement.
All equipment underwent a calibration procedure before the experiment to ensure signal accuracy.
The EEG device (Muse 2) was calibrated and attached by a trained assistant who checked the electrode contact status after Bluetooth synchronization, confirmed the stability of the reference electrode signal, and inspected external noise to maintain normal signal quality.
Before measuring HR, the research assistant checked the placement of the heart rate sensor (Polar H10) and verified Bluetooth synchronization. Each participant carried their own device during walking to ensure signal stability. The principal investigator supervised the entire process to ensure the accuracy and consistency of data collection.
(Revised section: 2.3. Methods (2.3.1 Methods of measurement), Lines (170-177)
However, although the portable EEG device used in this study has the advantage of being applicable in outdoor environments, it cannot be excluded that external factors such as weather conditions, illumination, and surrounding noise may have subtly influenced the measurements, unlike in laboratory settings.
This characteristic represents an inherent limitation of field studies conducted in natural environments, and we have stated that supplementary measurements under laboratory conditions will be necessary in future research.
(Revised section: 3.3. discussion, Lines (417-426)
4.Insufficient participant selection criteria and lack of information on physical activity experience
This study is not a clinical trial but a comparative study on the health-promoting effects of daily walking activities among middle-aged women living in urban areas.
The purpose of the study is not to intervene in specific treatments or diseases but to explore how mental and physical responses differ when performing ordinary walking activities in two different urban environments (school ground vs. forest trail).
Therefore, the study targeted general middle-aged women without physical limitations in walking activities, rather than individuals requiring medical control or clinical treatment, and all participants successfully completed the experiments.
The inclusion and exclusion criteria for participant selection were as follows:
â‘ Inclusion criteria: Middle-aged women aged 40–59 years living in urban areas.
â‘¡ Exclusion criteria: Individuals with physical disabilities or those who have difficulty walking more than 2km.
In addition, a preliminary questionnaire was used to assess participants’ habitual physical activity (weekly frequency and average duration) and their mental health status during the past two weeks (Table 2).
Table 2 presents information on participants’ physical activity frequency, exercise intensity, engagement in flexibility or resistance training, and psychological states such as stress, headache, and chest tightness.
All participants were healthy middle-aged women living in metropolitan areas, regularly engaging in walking activities and reporting relatively low levels of psychological stress.
These details have been supplemented and clearly described in the Participants section.
(Revised section: 2.2. Participants, Lines (131-149)
5. External factors
All experiments were conducted under consistent environmental conditions to minimize variables caused by environmental factors.
The sessions were carried out during the same week under clear weather conditions, between 10:00 a.m. and 5:00 p.m., with a temperature of 18–20 °C and wind speed of approximately 3 m/s.
Variables such as strong wind or rainfall were excluded in order to maintain environmental consistency.
(Revised section: 2.3. Methods(2.3.1 Methods of measurement), Lines (185-190)
However, long-term seasonal variations were not considered in this study, and we have suggested that future research should include comparative analyses by season and studies controlling environmental factors.
(Revised section: 4. Conclusion, Lines (185-190)
6. Information about participants' usual physical activity
In this study, participants’ habitual physical activity patterns were investigated through a preliminary questionnaire, and the results were included in Table 2 (General Characteristics of Participants).
It was confirmed that, on average, the participants were a healthy group with regular walking habits and low levels of stress and headache, and this information was reflected in both the Methods and Results sections.
(Revised section: 2.2. Participants, Lines (137-149)
In summary,
This study acknowledged the limitations of having no participant control group and a short-term experimental design, while providing additional explanations on equipment calibration procedures, participant characteristics, and environmental controls to enhance the reliability and clarity of the study.
In addition, each limitation item was explicitly described in the Limitations section and the Conclusion, and the future research directions—including control group design, long-term observation, and control of lifestyle variables—were specifically suggested.
Comments 6: Study limitations: single measurement of indicators, no consideration of seasonal factors, no study of long-term effects, and insufficiently detailed description of participant characteristics!!
Response 6: We agree with all the comments and have revised and supplemented the manuscript as follows:
This study is a preliminary, single-session study, making it difficult to verify long-term effects, and it has limitations in fully reflecting seasonal and climatic factors.
In addition, since participants’ lifestyle habits and psychological characteristics could not be completely controlled, we kindly acknowledge that future research should include long-term and multi-layered analyses involving various factors such as season, age, gender, and lifestyle.
Thank you very much.
(Revised section: 4. Conclusion, Lines (522-526)

Reviewer 3 Report
Comments and Suggestions for Authors
Thank you for your conscientious response to the reviewer's suggested edits and comments. The manuscript is much improved and provides important information and findings that will contribute to the next generation of studies.
Author Response
We sincerely appreciate your encouraging and thoughtful comments.
We are grateful that you recognized our efforts to revise the manuscript in response to the reviewers’ suggestions.
It is an honor to know that you found the revised version much improved and that the study provides meaningful findings that can contribute to future research in this field.
Your positive evaluation has been a great motivation for our research team. Thank you once again for your kind feedback and support.
Although this study has certain limitations—such as a relatively small sample size and the short-term nature of the observations—we fully recognize these constraints. We plan to conduct follow-up studies with a larger participant pool and more refined environmental control to expand and deepen our findings.
Thank you once again for your kind feedback and support.
